# Changes in Hand Hygiene Knowledge, Attitudes, and Practices Among Primary School Students: Insights from a Promotion Program in Guatemala

**DOI:** 10.3390/ijerph22030424

**Published:** 2025-03-14

**Authors:** Michelle Marie Pieters, Natalie Fahsen, Christina Craig, Kelsey McDavid, Kanako Ishida, Christiana Hug, Denisse Vega Ocasio, Celia Cordón-Rosales, Matthew J. Lozier

**Affiliations:** 1Center for Health Studies, Universidad del Valle de Guatemala, Guatemala City 01015, Guatemala; nfahsen@uvg.edu.gt (N.F.); ccordon@uvg.edu.gt (C.C.-R.); 2Centers for Disease Control and Prevention, Atlanta, GA 30329, USAngl7@cdc.gov (K.M.); kanakoishida@gmail.com (K.I.); cmariehug@gmail.com (C.H.); rhq1@cdc.gov (D.V.O.); wfu2@cdc.gov (M.J.L.); 3Epidemic Intelligence Service, Centers for Disease Control and Prevention, Atlanta, GA 30329, USA

**Keywords:** knowledge, attitudes, practices, hand hygiene, elementary schools, COVID-19

## Abstract

School-aged children are vulnerable to infectious diseases due to their developing immune systems and frequent social interactions. The COVID-19 pandemic underscored the importance of non-pharmaceutical interventions, like hand hygiene (HH). This study evaluated the changes achieved through a school-based intervention to Guatemalan primary school students’ HH knowledge, attitudes, and self-reported practices while collecting teacher feedback to inform future efforts. The intervention included handwashing festivals, environmental nudges, and the regular delivery of soap and alcohol-based hand rub (ABHR). Knowledge, attitudes, and practices (KAP) surveys were conducted pre- and post-intervention with 109 and 144 students, respectively. Six teachers participated in interviews to provide perspectives. Significant improvements were observed in students’ knowledge of HH’s role in preventing disease (pre: 84.4%; post: 96.5; *p* < 0.01) and recognition of critical moments (pre: 84.4%; post: 92.4%; *p* < 0.05). Self-reported practices also improved, with more students reporting washing their hands for 20 s or more (pre: 68.8%; post: 79.9%; *p* < 0.05). Fewer students reported liking ABHR after the intervention (pre: 89%; post: 78.5%; *p* < 0.05). Teachers reported increased HH practices and provided feedback to enhance interventions. These findings highlight the effectiveness of school-based interventions and emphasize the importance of addressing knowledge gaps and incorporating teacher insights for sustained public health benefits.

## 1. Introduction

School-aged children are at high risk of contracting and spreading respiratory infectious diseases, such as COVID-19, due to their immature immune systems and large amount of contact with other children at school [1]. These infections not only affect children’s health but can also have a negative impact on their educational outcomes due to frequent absences caused by illness [2]. The COVID-19 pandemic prompted several non-pharmaceutical interventions to curb viral transmission globally, including mask use, maintaining physical distance, restrictions on movement, and practicing proper hand hygiene [3]. Hand hygiene, defined as an action of hand cleansing by either washing the hands with soap and water or using an alcohol-based hand rub (ABHR), is a cost-effective preventative behavior for reducing the spread of disease [4]. However, global estimated rates of handwashing in community settings after using the restroom are only 19% [5]. A recent systematic review cited that knowledge, beliefs, and behaviors are some of the factors that affect hand washing in community settings [6].

The World Health Organization (WHO) has emphasized the important role that schools have in promoting health [7]. Schools play a crucial role in targeting hand hygiene behaviors and other health-related habits because children still have the capacity to develop new and lasting habits [6]. Educational interventions have the potential to increase hand hygiene and lead to significant health improvements, but previous studies have shown that effective behavior change requires more than just educational and informational campaigns [7]. Evidence suggests that environmental nudges, defined as subtle changes in the environment that promote desired behaviors, can be effective at changing hand hygiene behavior among school children [8].

The purpose of this study was to evaluate Guatemalan public primary school students’ hand hygiene knowledge, attitudes, and self-reported practices before and after an educational and environmental nudge intervention and to gather teacher feedback on the intervention to improve future implementation.

## 2. Materials and Methods

### 2.1. Study Population

The study was conducted between 2021 and 2023 and focused on 3rd- to 6th-graders from six public primary schools in three municipalities (San Miguel Sigüilá, San Juan Ostuncalco, and Concepción Chiquirichapa) of the Quetzaltenango department in Guatemala (Figure 1). Schools were selected by convenience based on Universidad del Valle’s (UVG) previous epidemiological surveillance work with the hospitals of the area and according to whether they were open for in-person classes during the COVID-19 pandemic (2021–2022).

### 2.2. Evaluation Methods

We evaluated educational and environmental nudge interventions using two different methods. We conducted knowledge, attitudes, and practices (KAP) surveys pre- and post-intervention with students to evaluate their understanding of hand hygiene. We conducted in-depth interviews with schoolteachers post-intervention with the aim of (1) gathering data concerning their perspectives on students’ hand hygiene practices, (2) assessing the intervention’s influence on students’ hand hygiene practices, and (3) seeking constructive feedback for improvement.

### 2.3. Study Participants

We included two groups of participants in the study: (1) students attending any of the six schools who were 8 years or older and (2) teachers who were involved in the implementation of the intervention and were willing to be audio-recorded. For the KAP surveys, an estimated 144 surveys (24 students per school) were needed at each time point to be able to detect a 10-percentage-point difference between pre- and post-intervention correct responses with 90% confidence. KAP surveys were carried out with a convenience sample of students from 3rd through 6th grades to avoid disrupting class time as much as possible (i.e., students had finished their classwork). KAP surveys were not paired pre- and post-intervention; we did not collect identifiable information and thus did not compare the responses of surveys conducted on the same students pre- and post-intervention. For the in-depth interviews, we selected one teacher from each school to participate through convenience sampling. Since the questions asked were specific to the intervention, we expected to reach thematic saturation within six interviews.

### 2.4. Intervention

The Manitas Limpias (Clean Little Hands) project aimed to improve the hand hygiene practices of primary school students in rural Guatemala through targeted interventions. The selection of interventions was primarily informed by three key sources: [1] findings from a baseline study that examined students’ hand hygiene knowledge, attitudes, and self-reported practices, [2] an assessment of the schools’ water, sanitation, and hygiene infrastructure, and [3] collaborative workshops involving teachers, principals, and officials of the Ministry of Education (MoE) of Guatemala [9,10].

The intervention consisted of three parts: (1) handwashing festivals, (2) environmental nudges, and (3) hand hygiene supply delivery. Handwashing festivals were conducted school-wide at all six participating schools with each school taking responsibility for the planning and execution of the event. Our team provided technical assistance to ensure that the festivals aligned with the project objectives. Each grade or class was assigned a specific aspect of hand hygiene to present to the rest of the school. The topics could range from critical moments for handwashing and steps for handwashing to germ transmission and the importance of hand hygiene. Schools had the liberty to decide how to present the information, whether through posters, dances, games, songs, puppet shows, or other creative methods aimed at communicating the information. Parents and other local educational authorities were invited to these festivals to strengthen community involvement.

Three main nudges were provided to the schools for implementation around the handwashing stations. First, schools were supplied with bright and colorful oil-based paint to create footstep markings leading from the toilets or playground to the handwashing stations (Figure 2). Additionally, step stools were provided to schools to place around the handwashing stations, facilitating access for smaller children to wash their hands (Figure 3). Lastly, in collaboration with teachers, principals, and officials from the MoE, a poster was designed containing information on the proper handwashing technique and placed above the handwashing stations (Figure 4).

To promote the sustainability of the intervention, each school was provided with a plan to hold handwashing festivals twice a year, reinforcing good hand hygiene habits over time. Additionally, the high-quality materials used for the nudges, such as paint and stools, were selected to ensure that they remained functional and visible beyond the immediate intervention period.

### 2.5. Outcomes Measured

A KAP survey was administered pre- and post-intervention to assess changes in understanding, beliefs, and self-reported practices regarding hand hygiene. Correct responses for each of the questions in the survey were defined a priori. The knowledge section comprised questions focusing on the duration of handwashing, materials needed for proper handwashing, the importance of handwashing, and critical moments for handwashing. In this section, some questions had multiple correct responses, and, in those cases, we marked the question as correct when the student got at least one correct answer. The attitudes section had questions focusing on personal (e.g., “Do you think handwashing is important to prevent diseases?”) and family and friends’ beliefs around hand hygiene (e.g., “Do you see your family and friends washing their hands with soap and water?”). In this section, a positive response to the statement was considered correct. The practices section asked about self-reported behaviors (e.g., “When you wash your hands, how long does it take?”). Like the knowledge section, even if a question had multiple correct responses, we marked it as correct if the student got at least one correct answer.

The in-depth interviews with teachers focused on their perceptions of students’ hand hygiene behaviors before and after the intervention, the impact each specific part of the intervention had on the school and the students, and suggestions for future activities.

### 2.6. Study Timeline

The baseline KAP survey was conducted between May and June 2022 prior to the intervention (Figure 5). The outcomes served as one of the sources for developing our intervention. Following this, a collaborative workshop (Workshop 1) was held in October 2022 with teachers and principals to gather feedback on the proposed intervention and initiate planning for the handwashing festivals. The first set of handwashing festivals took place in November 2022. In February 2023, a feedback workshop (Workshop 2) was held with teachers and principals to evaluate the effectiveness of the initial festivals and integrate suggestions for the subsequent festivals. The second set of handwashing festivals was held in April 2023. The follow-up KAP survey and the in-depth interviews were conducted in July 2023 as a post-intervention evaluation. Soap and ABHR were delivered monthly from October 2022 to October 2023.

### 2.7. Data Collection and Analysis

KAP surveys were conducted in Spanish during the school day by two to three native Spanish-speaking trained enumerators who recorded responses using the REDCap electronic data capture tool on a tablet. Answer options were not shown or read aloud to participants. Each survey took approximately 15 min to complete.

Semi-structured interviews were conducted in Spanish by one member of the research personnel trained in qualitative data collection with one teacher at each school. Interviews were carried out in person, audio-recorded, transcribed verbatim, and translated to English for analysis. Descriptive summaries explored students’ demographic characteristics (age, sex, and grade) and answers to the KAP survey questions. For the KAP surveys, we calculated the number and percentage of participants who responded correctly to each question. We performed chi-square tests (or Fisher’s exact test, where *n* < 5) to assess for significant differences in number of students that responded correctly to each question pre-and post-intervention (significance at *p* < 0.05). Data were analyzed using STATA version 19 (StataCorp LLC: College Station, TX, USA).

In-depth interviews lasted anywhere from 20 to 30 min. A coding framework was developed using a combined inductive and deductive approach; initial codes were developed based on the interview questions, and additional codes were added as themes emerged. Two reviewers independently coded one transcript using MAXQDA 2022 software (VERBI Software, 2021) and met to discuss the consensus on the code application. Later, one researcher applied the revised codebook to all other transcripts. After all transcripts were coded, the same researcher wrote analytic memos summarizing key themes across the interviews.

### 2.8. Ethical Considerations

This study was reviewed and approved by the Center for Health Studies of the Universidad del Valle de Guatemala (Protocol No. 246-05-2021).

## 3. Results

### 3.1. Socio-Demographic Information of KAP Survey Participants

In total, 109 students participated in the first survey and 144 in the second survey, of which 48.6% and 45.1% were girls, respectively (Table 1). Participants’ grades were equally distributed on both survey time points. Only one statistically significant difference (*p* < 0.05) between participants in the first and second survey was found: a greater proportion of participants in the second survey were 11 years old.

### 3.2. Changes in Knowledge, Attitudes, and Self-Reported Practices

Of the five questions included in the knowledge section of the survey, three showed a significant difference in the number of participants who answered correctly following the intervention (Table 2). The percentage of students who knew what materials are needed for handwashing increased from 94.5% to 100% (*p* < 0.05). Students identifying correct reasons for handwashing increased from 84.4% to 96.5% (*p* < 0.01). In addition, knowledge of critical moments for handwashing increased from 84.4% to 92.4% (*p* < 0.05). There was no significant difference regarding the number of students who answered correctly questions related to how long you should wash your hands (53.5% to 61.1%; *p* > 0.05) and materials needed to wash your hands if your hands are visibly dirty (pre: 78.0%; post: 81.9%; *p* > 0.05).

Among the six questions included in the attitudes section of the survey, two showed a significant difference in the number of students who answered correctly following the intervention (Table 3). Specifically, students’ perceived attitudes towards whether their friends and family considered handwashing important showed a significant increase between the pre- (91.7%) and post-surveys (96.5%) (*p* < 0.05). Additionally, students were significantly less likely to report liking to clean their hands with ABHR after the intervention (78.5%) than before the intervention (87.2%) (*p* < 0.05). We found no significant difference in participants’ attitudes regarding hand hygiene’s importance in preventing diseases, handwashing importance to friends and family, or seeing friends and family wash their hands with soap and water or using ABHR and no difference in ease or difficulty with washing their hands at home and school.

Among the six questions included in the practices section of the survey, three showed a significant difference in the number of students who answered correctly following the intervention (Table 4). The proportion of correct answers significantly increased when the participants were asked what materials they used to wash their hands (pre: 82.6%; post: 93.8%; *p* < 0.01), being able to name at least one critical moment for handwashing (pre: 85.3%; post: 95.1%; *p* < 0.01), and handwashing duration (pre: 68.8%; post: 79.9%; *p* < 0.05).

### 3.3. Intervention’s Influence on Students’ Hand Hygiene Practices

We interviewed six teachers who played an important role in the development and implementation of the Manitas Limpias project. All teachers mentioned that handwashing practices had increased since the implementation of the project at their school. They mentioned seeing more students wash their hands after using the restroom and before eating. All teachers mentioned that the supply of soap and ABHR provided as part of the project was an important part of the intervention since the school no longer had to purchase its own soap and could allocate those resources to other priorities, and students had materials available to wash their hands all the time, which was not the case pre-intervention. However, many teachers noted that 100% hand hygiene adherence is hard to achieve because there will always be some students who forget to wash their hands. Additionally, teachers mentioned that children do not practice hand hygiene at home because parents do not practice the habit, which affects their practice at school.

*Participant* *5:* *“Mom and Dad don’t teach them to wash their hands after going to the bathroom, after touching things that are contaminated, so the child grows up in that environment”.*

*Participant* *1:* *“I have realized, because I live here in the municipality, is that it is because the hygiene measures, the hygienic measures, are not practiced at home. I have seen, let’s say, a farmer who goes to fumigate his potatoes. He goes to fumigate and takes his bag of bread and when he finishes fumigating without washing his hands—considering that he is handling poison—he starts to eat [the bread]. So, since in your house you don’t do it, they don’t worry”.*

Overall, teachers had positive experiences organizing, participating, and executing the festivals. All of them said that the students enjoyed the festivals, as they included hands-on activities where they could display their abilities and learn from their peers. Many said that since the festivals, they had noticed a change in students’ knowledge and/or practice. They noticed that students know more about hand hygiene and that other students have been more aware of their own practices and the practices of their peers. Some teachers also mentioned that this project has served as a personal reminder to practice hand hygiene.

*Participant* *3:** “Well, I try to wash my hands because I touch a lot of things here in the school. Also, before entering [class], I tell my students, “Let’s wash our hands, my children,” and then they wash their hands. If I am touching books and their things, I try to apply [ABHR] to avoid contamination because the truth is that children have become very sick, and if we do not practice cleanliness, we can get all the viruses in our mouths or noses, or in our eyes”.*

All schools used the paint provided to draw footpaths from bathrooms/playgrounds to handwashing stations. Half of the teachers mentioned seeing changes in student’s handwashing behavior based on the paint alone, and the other half mentioned that they had not observed a change in students’ behavior.


*Interviewer:*
* “And have you observed any change in students’ hand hygiene because of the little hands and feet they painted, or do they have no impact?”*


*Participant* *1:** “Yes, yes, indeed. In fact, as I say, maybe not 100%, but some do [wash their hands]. They look at the images and then act based on the images. It’s a reminder”.*


*Interviewer:*
* “And do you think students see those little steps, and in some way, does it impact whether they wash their hands or not?”*


*Participant* *2:** “No, no, no. I don’t think it does”*

Step stools were provided to the schools to improve the accessibility of the handwashing station for smaller children. Teachers mentioned that the smaller students who could not previously reach the station would not wash their hands, but now they do. Teachers had a positive review of the designed poster that was placed above the handwashing stations. They mentioned seeing students interact with the material by stopping what they were doing, reading what was on the poster, and using it as a guide to washing their hands.


*Interviewer:*
* “Have you seen if there is any change in student hand hygiene because of the stools?”*


*Participant* *2:** “The truth is yes because I see that small children, who used to not be able to wash their hands, sometimes would only wash their little fingers. Now, they reach. They pull the stool, stand up, and wash their hands properly, as it should be”*

Teachers provided valuable insights on ways to enhance the intervention’s effectiveness, such as having more time to plan the handwashing festivals. They recommended scheduling these events when there are no competing school activities or holidays to maximize participation. There was also a consensus among teachers on the importance of including preschool in all project activities. Additionally, teachers suggested conducting the intervention activities in smaller groups, rather than at the school level. They expressed that this approach would be more manageable, especially for larger schools, and more in line with the developmental needs of the students.

## 4. Discussion

In this study, we aimed to evaluate the school-based hand hygiene intervention on students’ hand hygiene knowledge, attitudes, and practices in six primary schools in Guatemala. Our findings revealed positive changes in hand hygiene knowledge; specifically, there was a statistically significant change from pre- to post-intervention in correct responses relating to the importance of hand hygiene and recognizing at least one critical moment for handwashing. Moreover, there was a change in attitudes towards hand hygiene. However, there was also a negative change in students’ preference for using ABHR. Self-reported practices also showed positive changes, including identifying critical moments for handwashing, and improved adherence to washing their hands for 20 s or more.

These findings align well with previous research highlighting the effectiveness of a combined intervention (educational + nudges) in promoting hand hygiene behaviors in schools. In a cluster-randomized trial carried out in schools in Bangladesh, a nudge intervention was comparable to a high-intensity hygiene education on sustained improvements in handwashing behaviors five months after the interventions [11]. Previous research has also shown that education and training are important for increasing knowledge about hand hygiene but have not been effective at improving compliance; therefore, alternative or combined approaches to influencing behavior could show promise [12]. A systematic review looking at hand hygiene nudges in schools against the spread of COVID-19 found that nudges should be included in any public health intervention due to their positive effect on hand hygiene [13]. Specifically, studies using floor markings, like the footpaths leading to a hand hygiene station, have been shown to reduce barriers and facilitate access to hand hygiene [11,14]. In this study, however, some teachers perceived that the footpaths did not impact hand hygiene, highlighting the need to further research the impact of this nudge. Using posters that combine images with targeted messages has also acted as an effective reminder for people who tend to forget to perform hand hygiene [15,16]. Our poster was developed in collaboration with the teachers and principals from the school, as well as the Ministry of Education in Guatemala, to ensure that it included an appropriate message and culturally pertinent images, which could have improved its impact on students.

Interestingly, in other studies in hospitals, the use of ABHR increased because it was easily accessible to users [17]. However, in our study, we found the opposite. Even though schools were provided with enough ABHR to supply the whole school, and it was placed near the handwashing stations, students reported using it less and preferring to wash their hands with soap and water. Our study did not explore the reasons behind this decrease, but other studies have cited unpleasant ABHR residue and reports that it made the hands dry and irritated [18]. Additionally, cultural factors may influence students’ preferences. In some Mesoamerican cultures, including where this study took place, people often classify concepts or objects as having “hot” or “cold” properties that are not necessarily related to temperature. Some students mentioned not wanting to use ABHR, as it felt “hot,” and did not want to apply it to their “cold” hands [19].

The insights provided by teachers offer valuable guidance for future intervention efforts. Their feedback underscores the importance of planning and collaboration in intervention design. Although scheduling was up to the schools, we did provide guidance on what dates would work best for our study timeline. This caused some scheduling concerns, as schools already had lesson plans established and had to shift programs around to accommodate the festivals. Future handwashing festivals could be integrated into the year’s lesson plan to avoid competition with other school activities. Additionally, for our study, we did not consider preschool students, as not all schools included a preschool, and our study activities were targeted towards elementary-level students. However, interviewees indicated the importance of including younger students in these interventions, as hand hygiene habits can be formed as early as the age of 3 years [20].

To our knowledge, this study was the only one of its kind in school communities in Guatemala during COVID-19. Another strength of our study lies in our mixed-methods approach, which allowed for a comprehensive evaluation of the intervention and teacher perspectives. However, certain limitations should be acknowledged. First, we used a convenience sample to select schools and students who participated in the KAP surveys. This could impact the generalizability of our findings, as the representativeness of these schools beyond the local region cannot be assumed. The sampling frame of KAP survey participants and the data collection strategy did not allow for the pairing of students’ responses pre- and post-intervention. We did not conduct a stratified analysis to determine whether differences in correct answers differed by demographic characteristics. Additionally, relying on self-reported data could lead to a social desirability bias and students responding to what we wanted to hear, especially post-intervention, when they recognized our study team and our work.

Future research to explore the sustainability of hand hygiene behavior change beyond the immediate post-intervention period could enhance the understanding of the impact of these types of interventions. Longitudinal studies could provide valuable insights into the persistence of hand hygiene practices over time and which types of interventions contribute to lasting behavior change. Additionally, measuring the interventions separately (education and nudges independently) could also better indicate which intervention has a stronger impact on hand hygiene knowledge, attitudes, and practices.

## 5. Conclusions

In conclusion, the findings of this study highlight the positive change that an educational and environmental nudge intervention can have in promoting hand hygiene knowledge, attitudes, and self-reported practices among primary school students in Guatemala. By addressing the knowledge gaps, shaping attitudes, and reinforcing positive practices, these interventions can achieve public health benefits. By incorporating teacher feedback and considering recommendations for improvement, future interventions could have a larger impact on student health and wellbeing

## Figures and Tables

**Figure 1 ijerph-22-00424-f001:**
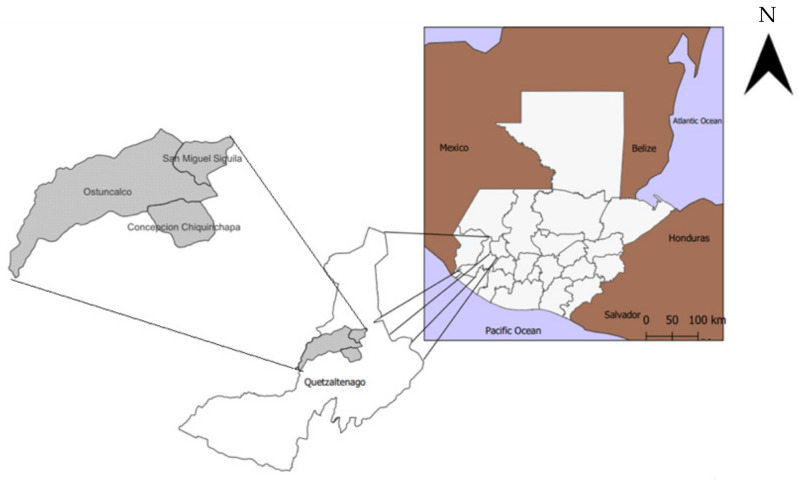
Map of study locations.

**Figure 2 ijerph-22-00424-f002:**
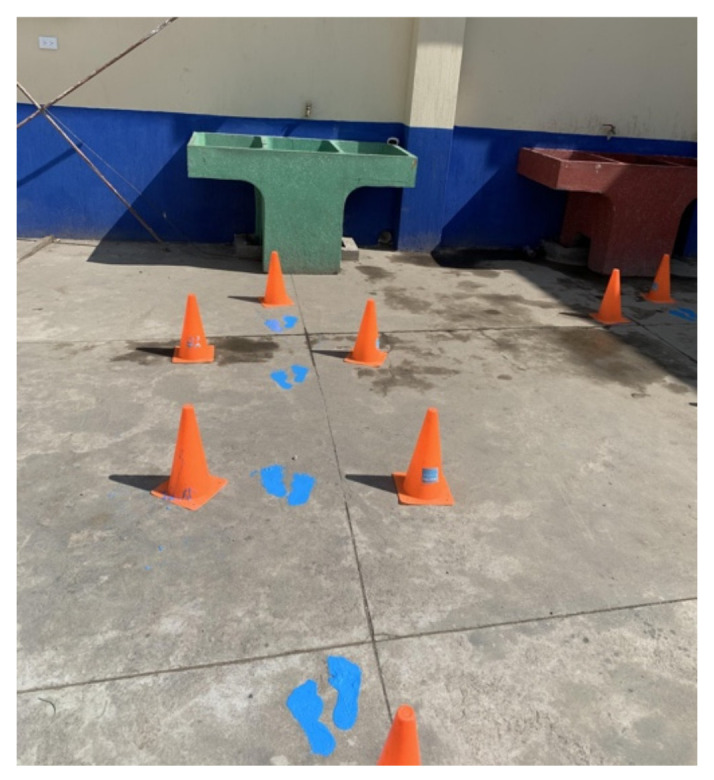
Footsteps from playground to handwashing station.

**Figure 3 ijerph-22-00424-f003:**
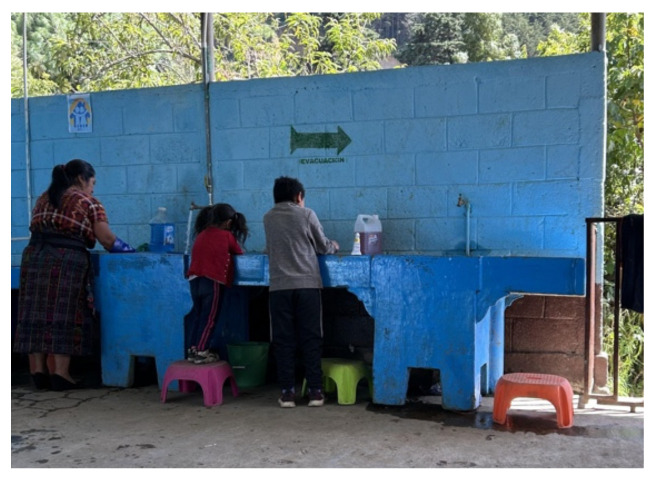
Step stools.

**Figure 4 ijerph-22-00424-f004:**
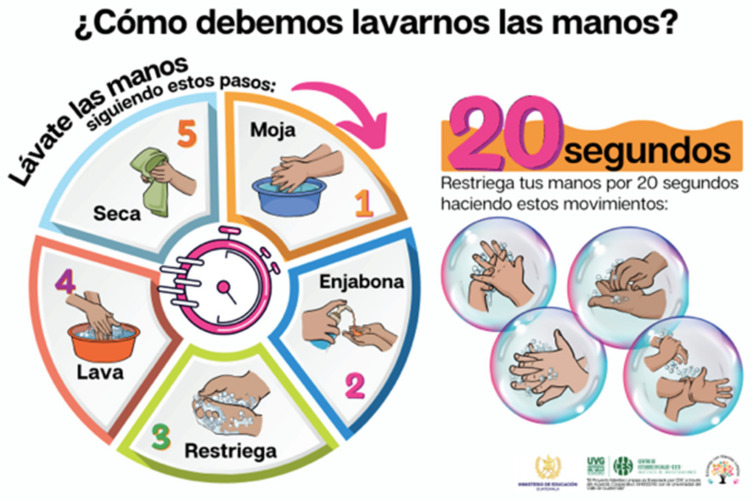
Poster design in collaboration with MoE. How should we wash our hands? Circle: Wash your hands following these steps: 1. Get them wet. 2. Get soap. 3. Lather them. 4. Rinse them. 5. Dry them. 20 S. Wash your hands for 20 S making these movements.

**Figure 5 ijerph-22-00424-f005:**
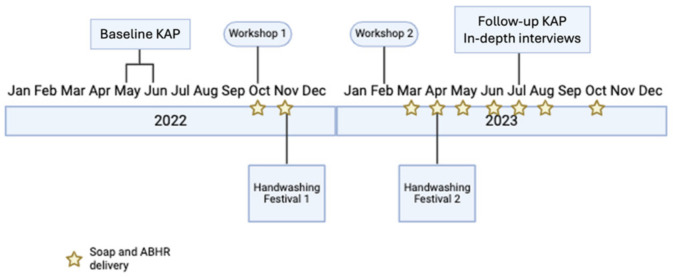
Study timeline. Note that schools are not in session December–February, so HH materials were not delivered during that time.

**Table 1 ijerph-22-00424-t001:** Participant demographics pre- and post-intervention.

	Pre-Intervention [*n* = 109] *n* (%)	Post-Intervention [*n* = 144] *n* (%)
Age		
8	10 (9.2)	5 (3.5)
9	24 (22.0)	29 (20.1)
10	36 (33.0)	32 (22.2)
11 *	15 (13.8)	37 (25.7)
12	19 (17.4)	31 (21.5)
13–15	5 (4.6)	10 (7)
Sex		
Male	56 (51.4)	79 (54.9)
Female	53 (48.6)	65 (45.1)
Grade		
3rd	27 (24.8)	35 (24.3)
4th	35 (32.1)	34 (23.6)
5th	27 (24.8)	39 (27.1)
6th	20 (18.3)	36 (25.0)

*: Significant variation between pre and post intervention group (significance at *p* < 0.05).

**Table 2 ijerph-22-00424-t002:** Changes in hand hygiene knowledge pre- and post- intervention.

Knowledge Question	Correct Response	Participants Who Responded Correctly		Percentage Point Difference *p*-Value
		Pre-Intervention [*n* = 109] *n* (%)	Post-Intervention [*n* = 144] *n* (%)	
For how many seconds do you think you should wash your hands?	20 s or more	58 (53.2)	88 (61.1)	+7.9 *p* = 0.21
What materials are needed for hand washing?	Water and soap or alcohol-based hand rub	103 (94.5)	144 (100)	+5.5 *p* < 0.05
Why is hand washing important?	To avoid getting sick or to prevent the transmission of germs/bacteria or to remove dirt	92 (84.4)	139 (96.5)	+12.1 *p* < 0.01
If your hands are dirty, what materials should you use to wash them?	Water and soap	85 (78.0)	118 (81.9)	+3.9 *p* = 0.43
When should you wash your hands?	Before eating or after coughing or sneezing or after going to the bathroom or after touching something dirty or after playing outside or after eating	92 (84.4)	133 (92.4)	+8 *p* < 0.05

**Table 3 ijerph-22-00424-t003:** Changes in hand hygiene attitudes pre- and post- intervention.

Attitudes Question	Correct Response	Participants Who Responded Correctly		Percentage Point Difference *p*-Value
		Pre-Intervention [*n* = 109] *n* (%)	Post-Intervention [*n* = 144] *n* (%)	
Do you think hand washing is important to prevent diseases?	Yes	104 (95.4)	140 (97.2)	+1.8 *p* = 0.66
Do your friends and family think hand washing is important?	Yes	100 (91.7)	139 (96.5)	+4.8 *p* = 0.10
Do you see your friends or family washing their hands with soap and water?	Yes	106 (97.2)	140 (97.2)	0 >0.99
Do you see your friends or family cleaning their hands with alcohol gel?	Yes	95 (87.2)	116 (80.6)	−6.6 *p* = 0.16
Do you like to wash your hands with soap and water?	Yes *	109 (100)	144(100)	No variance
Do you like to clean your hands with alcohol gel?	Yes *	97 (89)	113 (78.5)	−10.5 *p* < 0.05
When you are at school and you want to wash your hands, is it easy or difficult?	Easy	104 (95.4)	138 (95.8)	+0.4 >0.99
When you are at home and you want to wash your hands, is it easy or difficult?	Easy	107 (98.2)	139 (96.5)	−1.7 *p* = 0.70

* ”Yes” was the response used to measure correct answers.

**Table 4 ijerph-22-00424-t004:** Changes in self-reported practices pre- and post-intervention.

Practice Questions	Correct Response	Practices (% of Correct Self-Reported Practices)		Percentage Point Difference *p*-Value
		Pre-Intervention [*n* = 109] *n* (%)	Post-Intervention [*n* = 144] *n* (%)	
Have you washed your hands today?	Yes	98 (89.9)	137 (95.1)	+5.2 *p* = 0.11
What did you use to wash your hands?	Soap and water or alcohol-based hand rub	90 (82.6)	135 (93.8)	+11.2 *p* < 0.01
Do you wash your hands at home?	Yes	105 (96.3)	133 (92.4)	−3.9 *p* = 0.19
Do you wash your hands at school?	Yes	98 (89.9)	129 (89.6)	−0.3 *p* = 0.93
When do you wash your hands?	Before eating or after coughing or sneezing or after going to the bathroom or after touching something dirty or after playing outside or after eating	93 (85.3)	137 (95.1)	+9.8 *p* < 0.01
When you wash your hands, how long does it take you to wash your hands?	20 s or more	75 (68.8)	115 (79.9)	+11.1 *p* < 0.05

## Data Availability

Data are available from the authors upon reasonable request.

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
