# Peer review of "Changes in Hand Hygiene Knowledge, Attitudes, and Practices Among Primary School Students: Insights from a Promotion Program in Guatemala"

_ijerph, 2025, doi:10.3390/ijerph22030424_

Round 1

Reviewer 1 Report

Comments and Suggestions for Authors

This paper summarizes the effects of an intervention on hand hygiene among primary school students. The content is very interesting, but the paper would be stronger if the intervention effects were made clearer.

Major points

1. It is clear that the attitudes and knowledge of primary school students regarding hand hygiene have improved, but did you evaluate whether they are actually practicing it effectively? For example, using fluorescent paint and UV lamps can help assess if there are any areas missed during hand washing. Inadequate hand hygiene will not be sufficient to prevent infections effectively.

2. The statement "To our knowledge, this study is the only of its kind in school communities in Guatemala during COVID-19" is mentioned, but were there any visible intervention effects, such as a reduction in the number of COVID-19 or other infectious disease cases, as a result of the hand hygiene intervention?

Minor points

1. It is common practice to include the figure legend at the bottom of each figure.  In particular, Figure 4 is written in a language other than English and requires a detailed explanation. Please provide a detailed description so that it is clear what specific hand hygiene practices were instructed.

2. I think it would be clearer to add subheadings to the Materials and Methods section.

Reviewer 2 Report

Comments and Suggestions for Authors

The study is relevant in assessing the effectiveness of a hand hygiene promotion program in primary schools in Guatemala. It is well-structured and follows an appropriate methodological design, combining quantitative (Knowledge, Attitudes, and Practices – KAP surveys) and qualitative (teacher interviews) approaches. The results indicate significant improvements in students' hand hygiene knowledge and practices following the intervention, although some barriers persist, such as inconsistent adherence and the influence of the home environment. The discussion is well-grounded and contextualized with relevant literature, highlighting the importance of school-based interventions in promoting public health. However, here are my considerations for improving the manuscript:

- Add the research ethics committee number in the study methodology.

- The results showed a decrease in students' preference for using alcohol-based hand rub. Therefore, please provide a more in-depth discussion of the reasons for this. 

- I suggest a more comprehensive discussion on the sustainability of the intervention implemented in the study (How can the habits be maintained after the project's conclusion? What is the role of parents and the community?).

- I suggest, if possible, inserting or changing references for more recent ones (last five years). Of the total 20 references in the study, only 45% (9) are from the last five years and there are no references from 2024 and 2025.

 In conclusion, the study is well-written, scientifically robust, and addresses a highly relevant topic: the importance of hand hygiene as a fundamental measure for controlling infectious diseases in daily life. Despite some methodological limitations discussed at the end of the manuscript, the findings are valuable and contribute to the knowledge on school-based public health interventions.

Round 2

Reviewer 1 Report

Comments and Suggestions for Authors

The author has appropriately revised the manuscript.